# Spatially-Guided Temporal Attention (SGuTA) and Shifted-Cube Attention (SCubA) for Video Frame Interpolation

## Abstract

In recent years, methods based on convolutional kernels have achieved state-of-the-art performance in video frame interpolation task. However, due to the inherent limitations of their convolutional kernel size, it seems that their performances have reached a plateau. On the other hand, Transformers are gradually replacing convolutional neural networks as a new backbone structure in image tasks, thanks to their ability to establish global correlations. However, in video tasks, the computational complexity and memory requirements of Transformer will become more challenging. To address this issue, we employ two different Transformers, SGuTA and SCubA, in VFI task. SGuTA utilizes the spatial information of each video frame to guide the generation of temporal vector at each pixel position. Meanwhile, SCubA introduces local attention into the VFI task, which can be viewed as a counterpart of 3D convolution in local attention Transformers. Additionally, we analyze and compare different embedding strategies and propose a more balanced embedding strategy in terms of parameter count, computational complexity, and memory requirements. Extensive quantitative and qualitative experiments demonstrate that our models exhibit high proficiency in handling large motions and providing precise motion estimation, resulting in new state-of-the-art results in various benchmark tests. The source code can be obtained at https://github.com/esthen-bit/SGuTA-SCubA.

## 1 Introduction

Video frame interpolation (VFI) is the process of reconstructing uncaptured intermediate frames during the exposure time by synthesizing adjacent frames, which can enhance its visual quality and smoothness of motion. As a fundamental problem in computer vision, it requires an understanding of both spatially and temporally consistence within the video frames, breaking the limitations of video sampling rate and lighting conditions. Its applications span across diverse domains, including virtual reality [1], video compression [2, 3, 4], and slow-motion generation [5, 6].

The majority of state-of-the-art techniques for VFI rely on convolutional neural networks (CNNs), particularly those based on kernels [7, 8, 9, 10, 11, 12, 13, 14], which have gained increasing popularity in recent years. Nevertheless, due to the inherent constraint imposed by the kernel size, convolutional kernels seem to have reached their performance ceiling, even after undergoing transformations such as 2D kernels, separable kernels, deformable kernels, and 3D kernels. It appears that there is a limited potential for further improvement of VFI methods based on CNNs and their associated kernels. Meanwhile, Transformers [15] have recently demonstrated its great potential in various image tasks such as image classification [16, 17, 18], object detection [19, 20], spectral reconstruction [21], and image restoration [22, 23, 24], due to their ability to capture long-range

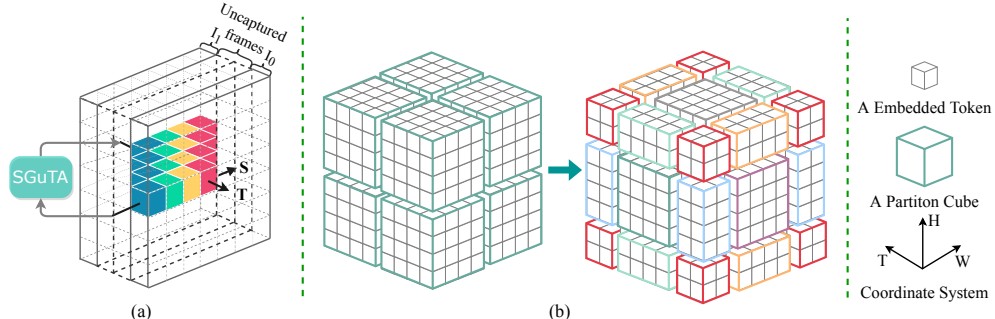

Figure 1: a) A simple illustration of the correlation between space and time within a video. The colored pixels move from left to right, these uncaptured frames can be restored because $\mathbf{T} = \mathbf{S}$. b) A simple illustration of shifted cubes approach, where the boundaries of the dimensions are connected, and the cubes with the same color are merged and masked after being shifted.

dependencies and contextual relationships in sequences. However, extending the Transformer to video tasks is not as straightforward as extending 2D convolutions to 3D convolutions, as it poses challenges such as computational complexity and memory requirements.

This article introduces two distinct Transformer-based approaches, SCubA (Shifted-Cube Attention) and SGuTA (Spatially-Guided Temporal Attention), which are integrated into a multi-stage multi-scale framework for VFI task. Both methodologies exhibit linear computational complexity with respect to the patch number, making them concise, efficient, and demonstrating exceptional performance.

It is revealed by [25] that there exists an inherent correlation between the spatial information and temporal sequence of a video. Fig. 1(a) illustrates a simple example for this phenomenon. If we exchange any spatial dimension (height or width) with the temporal dimension, a new video sequence can be obtained in which the low-resolution version of the original spatial information is recurred. Therefore, the higher-resolution original spatial information can provide powerful guidance for improving the temporal resolution. Inspired by this, we propose SGuTA, a self-attention mechanism that establishes intrinsic correlations between spatial information and temporal sequence.

Inspired by [16, 18, 26], as shown in Fig. 1(b), SCubA treats 3D-patches as tokens and partitions them into cubes with a fixed size along the height, width, and time axis. Local self-attention is computed within each cube, followed by shifted-cube mechanism to establish connections between adjacent cubes. This approach enables the model to exploit spatiotemporal locality inductive bias and achieve better performance than existing methods.

The main contributions of this work are listed as follow:

1) We present a novel Transformer called SGuTA, which is designed to establish the inherent correlations between the spatial characteristics and the temporal sequence within a video. SGuTA outperforms VFIT-B [14] in terms of PSNR by 0.58dB on vimeo-90k test set.

2) We propose a method called SCubA, which applies Video Swin Transformer [26] to VFI task. Compared to VFIT-B, SCubA achieves a PSNR improvement of 1.08dB while reducing both the number of parameters (Params) and computational complexity (FLOPs) by approximately 40%.

3) We conduct a analysis of existing embedding strategies, and put forth a novel half-overlapping embedding strategy. This method exhibits a more balanced performance in relation to Params, computational complexity, and memory usage.

## 2 Related Works

### 2.1 Video Frame Interpolation

The objective of VFI is to generate intermediate frames by combining adjacent frames that were not captured during the exposure period. This longstanding and classical problem in video processing is currently tackled through three prominent approaches: phase-based methods, optical flow-based methods, and kernel-based methods.

**Phase-based methods** [27, 28] utilize Fourier theory to estimate motion by analyzing the phase discrepancy between corresponding pixels in consecutive frames. These techniques generate intermediate frames by applying phase-shifted sinusoids. However, the $2\pi$-ambiguity problem can pose a significant challenge in determining the correct motion.

**Flow-based methods** [5, 10, 29, 30, 31, 32, 33] utilize optical flow estimation to perceive motion information and capture dense pixel correspondence between frames. These methods use a flow prediction network to compute bidirectional optical flow that guides frame synthesis, along with predicting occlusion masks or depth maps to reason about occlusions. However, these methods are limited by the accuracy of the underlying flow estimator remaining challenging problems in real-world videos, especially when there is large motion and heavy occlusion.

**Kernel-based methods** have gained momentum in VFI since the emergence of AdaConv [7], a method that uses a fully convolutional network to estimate spatially adaptive convolution kernels. This is because it no longer requires motion estimation or pixel synthesis like flow-based methods. DSepConv [9] and AdaCoF [10] employ Deformable convolution to overcome the limitation of a fixed grid of locations in original convolution. CAIN [11] expands the receptive field size of convolution by utilizing Pixel Shuffle. SepConv [8] performs separable convolution, thereby reducing the Params and memory usage. Then, FLAVR [13] substitutes the 2D convolutions utilized in Unet with their 3D counterparts, while applying feature gating to each of the resultant 3D feature maps. This achieves the best performance among CNN-based methods at the cost of a large Params. However, these CNN-based architectures still cannot overcome their inherent limitation of using fixed-size kernels, which prevent them from capturing global dependencies to handle large motion and limit their further development for VFI task. Inspired by Depth-wise separable convolution [34], Zhihao Shi et al. introduce VFIT [14], a separated spatio-temporal multi-head self-attention mechanism, which outperforms all existing CNN-based approaches while significantly reducing the Params. Within the field of kernel-based methods, CNN backbones have undergone a developmental trajectory from 2D to separable and then to 3D kernels. Zhihao Shi et al. has proposed a space-time separation strategy [14] in Transformer methods. In this work, we introduce a 3D version of the local self-attention mechanism and a spatially-guided temporal self-attention mechanism to the VFI task.

## 2.2 Vision Transformer

The key innovation of the ViT [16] is its application of the Transformer architecture, originally developed for natural language processing, to computer vision tasks. This represents a notable departure from the standard backbone architecture of CNNs in computer vision. By dividing the image into a sequence of patches and leveraging the Transformer encoder to capture global dependencies between them, ViT achieves impressive performance on image classification benchmarks. This pioneering work has paved the way for subsequent research aimed at improving the utility of the ViT model, and underscores the potential of the Transformer architecture in computer vision applications. To mitigate the computational and memory challenges associated with ViT, Swin-Transformer [18] partitions the embedded patches into non-overlapping windows. Within each window, local self-attention is calculated by ViT. Subsequently, shifted-window self-attention is computed to establish the correlation among windows. This strategy has demonstrated remarkable performance in various image tasks, such as image classification [18, 35], object detection [20, 19], and image restoration [36, 24], achieving state-of-the-art results. Despite Swin-Transformer's success in image tasks, extending it to video tasks by simply expanding along the time dimension resurfaces thorny computational and memory issues [14]. To address these issues, Ze Liu et al. further proposed a new 3D shifted windows mechanism [26] that efficiently captures temporal information, reduces the computational and memory demands, and achieves state-of-the-art results on video action recognition tasks. This method makes Swin-Transformer a promising approach for video analysis tasks.

# 3 Proposed method

SCuBA and SGuTA share this same network architecture. Fig. 2(a) depicts a multi-stage architecture that utilizes $N_s$ cascaded Multi-Scale Transformer. Instead of selecting two or four adjacent frames next to the reference frame $I_{0.5}$ as input, as in previous methods [11, 13, 14], our approach chooses six adjacent frames to accurately estimate the motion of the interpolated frame. Moreover, a long identity mapping is employed to mitigate the vanishing gradient problem. The desired interpolated

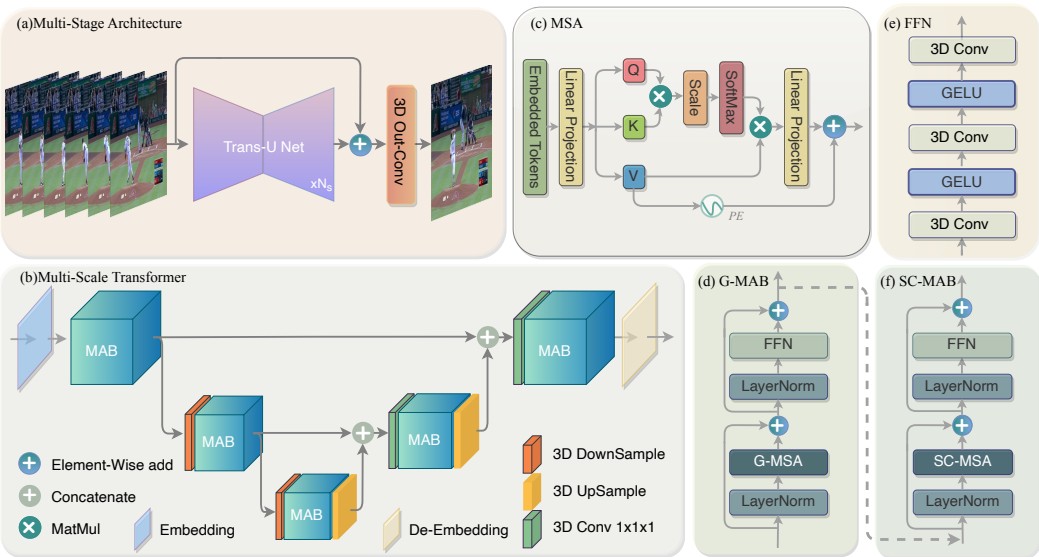

Figure 2: The overall pipeline of SGuTA and SCubA. a) Multi-stage Architecture. b) Multi-scale Transformer. c) Brief explanation of Multi-head Self-Attention. d) Global Multi-head Self-Attention Block (G-MAB). e) Feed Forward Network. f) Shifted-Cube Multi-head Self-Attention Block (SC-MAB).

frame $\hat{I}_{0.5}$ is finally obtained via a 3D convolution operation. Fig. 2(b) illustrates the network structure of the Multi-Scale Transformer when $N_s = 1$. Specifically, in the embedding layer, patches of frames are transformed into dense representations. The de-embedding layer performs the inverse operation of the embedding layer, whereby the representations are restored to patches. To enable multi-scale self-attention, it is essential to downsample the output of the Multi-Head Attention Block (MAB) from the previous scale before each MAB layer in the encoder. Similarly, in the decoder, the upsampling of the output of each MAB layer is first performed to restore the original spatial resolution, before sending into the next scale. Moreover, skip connections are employed at same scale, while a $1 \times 1 \times 1$ convolutional operation is applied to halve the depth of the concatenated feature maps.

SGuTA and SCubA are two transformer-based models that differ in their self-attention mechanisms. SGuTA is derived from the global self-attention mechanism, and its MAB module comprises solely the G-MAB module illustrated in Fig. 2(d). In contrast, SCubA is based on the local self-attention mechanism and its MAB module is constituted by the G-MAB module shown in Fig. 2(d), as well as the SC-MAB module depicted in Fig. 2(f), with the dotted line being solely applicable when utilizing SCubA. The composition of the feed-forward network (FFN) is presented in Fig. 2(e), whereas a concise procedure of multi-head self-attention (MSA) is portrayed in Fig. 2(c) (with some details omitted for concision). The disparity between SCubA and SGuTA is situated in the MSA module, which will be expounded upon in Section 3.1

## 3.1 Proposed MSA

### 3.1.1 SGuTA

Assuming an input tensor of shape $X_{in} \in \mathbb{R}^{T \times H \times W \times D}$, where $D$ denotes the length of embedding vector , the MSA module of SGuTA first transposes and reshapes it into a 2D tensor $X \in \mathbb{R}^{HW \times TD}$. This reshaped tensor $X$ is then projected through linear transformations $W_Q$, $W_K$, and $W_V \in \mathbb{R}^{TD \times TD}$ to obtain the query $Q$, key $K$, and value $V \in \mathbb{R}^{HW \times TD}$, respectively:

$$Q = XW_Q, K = XW_K, V = XW_V \tag{1}$$

Such a transformation enables MSA to leverage the interactions among different spatial features within the input tensor, facilitating the capturing of complex dependencies in the subsequent processing. Then, $Q, K, V \in \mathbb{R}^{HW \times TD}$ are divided into $n$ heads: $Q = [Q_1, ..., Q_n]$, $K = [K_1, ..., K_n]$,

$V = [V_1, ..., V_n]$, so that each head has a dimension of $d_h = \frac{TD}{n}$. The remaining process of SGuTA can be expressed as follows:

$$SGuTA(Q_i, K_i, V_i, d) = [\underset{j=1}{\overset{n}{Concat}}(head_j)]W + P(V), head_j = V_j softmax(\frac{Q_j^T K_j}{d}) \quad (2)$$

where $d \in \mathbb{R}^1$ and $W \in \mathbb{R}^{TD \times TD}$ are learnable parameters. $P(V) = 3DConv(Gelu(3DCon(v)))$ to generate positional embedding. The output $X_{out} \in \mathbb{R}^{T \times H \times W \times D}$ are obtained by reshaping the result of Eq. (2). Observing that SGuTA establishes correlations from space to time. Compared to the Global MSA method of establishing spatio-temporal correlations between all patches, SGuTA is capable of effectively alleviating memory requirements and computational complexity issues. The computational complexity of SGuTA can be easily obtained as follows:

$$\Omega(SGuTA) = 4TD^2(THW) + \frac{2TD^2}{n}(THW) \quad (3)$$

### 3.1.2 SCubA

In accordance with [26], the input tensor $X_{in} \in \mathbb{R}^{T \times H \times W \times D}$ is subjected to a process of partitioning into $\frac{THW}{thw}$ non-overlapping cubes of size $t \times h \times w$ utilizing an even partitioning strategy, as presented in Fig. 1b. The resulting cubes are reshaped into $x \in \mathbb{R}^{thw \times D}$ by the MSA module of SCubA. Linear transformations, specifically $w_q$, $w_k$, and $w_v \in \mathbb{R}^{D \times D}$, are employed to produce the query $q$, key $k$, and value $v \in \mathbb{R}^{thw \times D}$ representations, respectively.

$$q = xw_q, k = xw_k, v = xw_v \quad (4)$$

Similarly, $q, k, v \in \mathbb{R}^{thw \times D}$ are divided into $n$ heads: $q = [q_1, ..., q_n]$, $k = [k_1, ..., k_n]$, $v = [v_1, ..., v_n]$, so that each head has a dimension of $d_h = \frac{D}{n}$. The multi-head self-attention operation is then conducted within each cube according to the following equation:

$$SCubA(q_i, k_i, v_i, d) = [\underset{j=1}{\overset{n}{Concat}}(head_j)]W + P(v), head_j = softmax(\frac{q_j k_j^T}{d})v_j \quad (5)$$

To establish connections among the cubes, each cube is shifted along the time, height, and width dimensions by $t/2$, $h/2$, and $w/2$ steps, respectively, as depicted in Fig. 1b. The SC-MSA (corresponding to Fig. 2f) is calculated within each new cube.

The process in Eq. (4) and Eq. (5) is calculated for $\frac{THW}{thw}$ times, and its computational complexity can be specifically expressed as:

$$\Omega(SCuBA) = 4D^2(THW) + 2thwD(THW) \quad (6)$$

### 3.2 Other MSAs

In general, Global MSA [16] and Feature MSA [21] follow a standard procedure: the input $X_{in} \in \mathbb{R}^{T \times H \times W \times D}$ is reshaped and linearly transformed using $W_Q'$, $W_K'$, and $W_V' \in \mathbb{R}^{D \times D}$ to obtain $Q'$, $K'$, and $V' \in \mathbb{R}^{THW \times D}$, which are then divided into $n$ heads. Specifically, $Q' = [Q_1', ..., Q_n']$, $K' = [K_1', ..., K_n']$, and $V' = [V_1', ..., V_n']$, with each head having a dimension of $d_h = \frac{D}{n}$.

For Global MSA and Feature MSA, The multi-head self-attention is obtained by:

$$Global(Q_j', K_j', V_j', d) = [\underset{j=1}{\overset{n}{Concat}}(head_j)]W' + P(V'), head_j = softmax(\frac{Q_j' K_j'^T}{d})V_j' \quad (7)$$

$$Feature(Q_j', K_j', V_j', d) = [\underset{j=1}{\overset{n}{Concat}}(head_j)]W' + P(V'), head_j = V_j' softmax(\frac{Q_j'^T K_j'}{d}) \quad (8)$$

The computational complexity for Global MSA and Feature MSA is respectively given by:

$$\Omega(Global) = 4D^2(THW) + 2D(THW)^2 \quad (9)$$

$$\Omega(Feature) = 4D^2(THW) + \frac{2D^2}{n}(THW) \quad (10)$$

We validate the performance of the MSAs listed above, in addition to STS and Sep-STS [14], in the VFI task. Specific results and analysis can be found in Section 4.3.2.

### 3.3 Half Overlapping Embedding Strategy

We observe that the embedding strategies of Transformers can be mainly classified into two categories: the Non-overlapping [16] and Wide-overlapping [14] embedding strategy, which have no overlap and significant overlap between adjacent patches respectively. Specifically, if the patch size is set to $t \times h \times w$, the Non-overlapping and Wide-overlapping embedding strategies extract patches with strides of $t \times h \times w$, and $1 \times 1 \times 1$ respectively. Clearly, on the one hand, different stride will significantly impact the number of tokens and further affect the memory requirements during training. On the other hand, the length of the representation will change Params. Both factors influence the final performance of the model. We found that different embedding strategies maintain comparable performance when the following equation is satisfied:

$$\frac{D_2}{D_1} = \sqrt{\frac{t_1 h_1 w_1}{t_2 h_2 w_2}} \tag{11}$$

Here, $D_1$ and $D_2$ respectively represent the length of the embedding representation when patches are extracted with strides of $t_1 \times h_1 \times w_1$ and $t_2 \times h_2 \times w_2$.

Hence, we propose a compromise solution - the Half-overlapping embedding strategy - where adjacent patches overlap by half of their area or the stride is set to $t \times h/2 \times w/2$. The length of its embedding representation is set by Eq. (11). A detailed performance comparison and analysis of the three different embedding strategies can be found in Section 4.3.1.

## 4 Experiment

### 4.1 Implementation Details

**Training:** Consistent with [13], a basic $l_1$ loss is employed to train the networks: $||I_{0.5} - \hat{I}_{0.5}||$. The training batch size is set to 4, and the cube size of SCubA is set to $2 \times 4 \times 4$. The Adam optimizer [37] is utilized with $\beta_1 = 0.9$ and $\beta_2 = 0.99$. The learning rate is initialized to $2e^{-4}$, and a Cosine Annealing scheme is adopted over 100 epochs. Both SGuTA and SCubA employ Half-overlapping strategy with patch size setting to $1 \times 4 \times 4$ pixels.

**Dataset:** In this study, we use the Vimeo-90K septuplet training set [38] for training, it includes 64,612 seven-frame sequences with a resolution of $448 \times 256$. We selected the middle frame from each sequence as the ground truth, and pad one blank frame to the beginning and end of the remaining six frames. After random cropping, we obtain a video sequence of size $8 \times 3 \times 128 \times 128$ as input. We use the data augmentation method of FLAVR [13], which randomly applied horizontal and vertical flips and temporal flips to the input video sequence.

The performance of our models is accessed on widely-used datasets, including the Vimeo-90K septuplet test set [38], which comprises 7824 septuplets with a resolution of $448 \times 256$; the DAVIS dataset [39], containing 2849 triplets with a resolution of $832 \times 448$; and the SNU-FILM dataset [11], which is classified four categories based on the degree of motion: Easy, Medium, Hard, and Extreme. Each category comprises 310 triplets, primarily with a resolution of $1280 \times 720$. we transform the DAVIS dataset and SNU-FILM dataset into septuplets while preserving the ground truth to accommodate our network requirements and ensure fairness in comparing various models.

### 4.2 Evaluation against the State of the Arts

We conducted a comparative analysis of SGuTA and SCubA with competitive state-of-the-art methods, including SuperSlomo [31], SepConv [8], QVI [30], BMBC [32], CAIN [11], AdaCoF [10], FLAVR [13], VFIT-S [14], VFIT-B [14]. Tab. 1 reports the performance of each model in terms of Params, FLOPs, peak signal-to-noise ratio (PSNR), and structural similarity index (SSIM) on the Vimeo-90K and Davis datasets. Compared with the current SOTA method VFIT on the Vimeo-90K dataset, SGuTA achieves a significant performance improvement of **0.58dB** with similar Params and FLOPs. Moreover, SCubA reduces the Params and FLOPs by 40% and 39%, respectively while achieving a notable performance improvement of **1.08dB**. Fig. 3 illustrates the PSNR-FLOPs-Params comparison of these methods, which demonstrated that both SCubA and SGuTA are located in the upper-left region of the figure.

Table 1: Quantitative comparisons on the Vimeo-90K and DAVIS datasets.

| Methods | Params (M) | FLOPs (G) | Vimeo-90K | DAVIS |
|---|---|---|---|---|
| SuperSloMo [31] | 39.61 | 49.81 | 32.90/0.957 | 25.65/0.857 |
| SepConv [8] | 21.68 | 25.00 | 33.60/0.944 | 26.21/0.857 |
| QVI [30] | 29.21 | 72.93 | 35.15/0.971 | 27.17/0.874 |
| BMBC [32] | 11.01 | 175.27 | 34.76/0.965 | 26.42/0.868 |
| CAIN [11] | 42.78 | 43.50 | 34.83/0.970 | 27.21/0.873 |
| AdaCoF [10] | 21.84 | **24.83** | 35.40/0.971 | 26.49/0.866 |
| FLAVR [13] | 42.06 | 133.14 | 36.30/0.975 | 27.44/0.874 |
| VFIT-S [14] | **7.54** | 40.09 | 36.48/0.976 | 27.92/0.885 |
| VFIT-B [14] | 29.08 | 85.03 | 36.96/0.978 | 28.09/0.888 |
| **SGuTA** | 27.60 | 73.55 | 37.54/0.980 | 28.39/0.892 |
| **SCubA** | 17.30 | 51.71 | **38.04/0.981** | **28.86/0.899** |

Figure 3: PSNR-FLOPS-Params comparisons on Vimeo-90K dataset

Tab. 2 reports the performance of each model on the SNU-FILM dataset. Compared with the third-best model, SGuTA and SCubA achieve an average improvement of 0.67dB and 0.96dB, respectively, and a remarkable improvement of **1.14dB** and **1.59dB** in Hard scenario. This indicates that SGuTA and SCubA fully utilize the Transformer's ability to establish long-range correlations and prove their capability to handle challenging large-motion scenarios.

We provide qualitative results comparing our SGuTA and SCubA models to FLAVR [13] and VFIT [14]. As shown in Fig. 4. The first two rows fully demonstrate the ability of SGuTA and SCubA to provide accurate motion estimation (please carefully compare the rotation of the wheels and balls with the ground truth; other methods fail to restore the accurate rotation angles). The third row shows the performance of various models in non-rigid motion scenarios, where only SCubA clearly restores all the letters. In the fourth row, SGuTA and SCubA reconstruct clearer texture details. The last two rows again demonstrate the strong ability of our models to handle large motion scenarios.

Table 2: Quantitative comparisons on the SNU-FILM datasets.

| Methods | SNU-FILM | | | |
|---|---|---|---|---|
| | Easy | Medium | Hard | Extreme |
| SuperSloMo [31] | 37.28/0.986 | 33.80/0.973 | 28.98/0.925 | 24.15/0.845 |
| SepConv [8] | 39.41/0.990 | 34.97/0.976 | 29.36/0.925 | 24.31/0.845 |
| BMBC [32] | 39.88/0.990 | 35.30/0.977 | 29.31/0.927 | 23.92/0.843 |
| CAIN [11] | 39.92/0.990 | 35.61/0.978 | 29.92/0.929 | 24.81/0.851 |
| AdaCoF [10] | 40.08/0.990 | 35.92/0.980 | 30.36/0.935 | 25.16/0.860 |
| FLAVR [13] | 40.43/0.991 | 36.36/0.981 | 30.86/0.942 | 25.41/0.867 |
| VFIT-S [14] | 40.43/0.991 | 36.52/0.983 | 31.07/0.946 | 25.69/0.870 |
| VFIT-B [14] | 40.53/0.991 | 36.53/0.982 | 31.03/0.945 | 25.73/0.871 |
| **SGuTA** | 40.79/0.991 | 37.41/0.985 | 32.17/0.957 | 26.15/0.880 |
| **SCubA** | **40.90/0.992** | **37.78/0.986** | **32.62/0.960** | **26.37/0.884** |

## 4.3 Ablation Study

### 4.3.1 Embedding Strategy

In this section, we explore the relationship between various embedding strategies and the length of embedding vectors $D$. Taking SCubA as an example, given the input video size of $T \times H \times W = 8 \times 128 \times 128$, we set $N_s = 2$ and the patch size to $1 \times 4 \times 4$. The Wide, Half and Non-Overlapping Strategies extract patches with stride of $1 \times 1 \times 1$, $1 \times 2 \times 2$, $1 \times 4 \times 4$, respectively. As shown in Tab. 3, changing the Wide-Overlapping Strategy to the Non-Overlapping Strategy while keeping the size of $D = 32$ the same can reduce FLOPs and memory usage, but lower the performance.

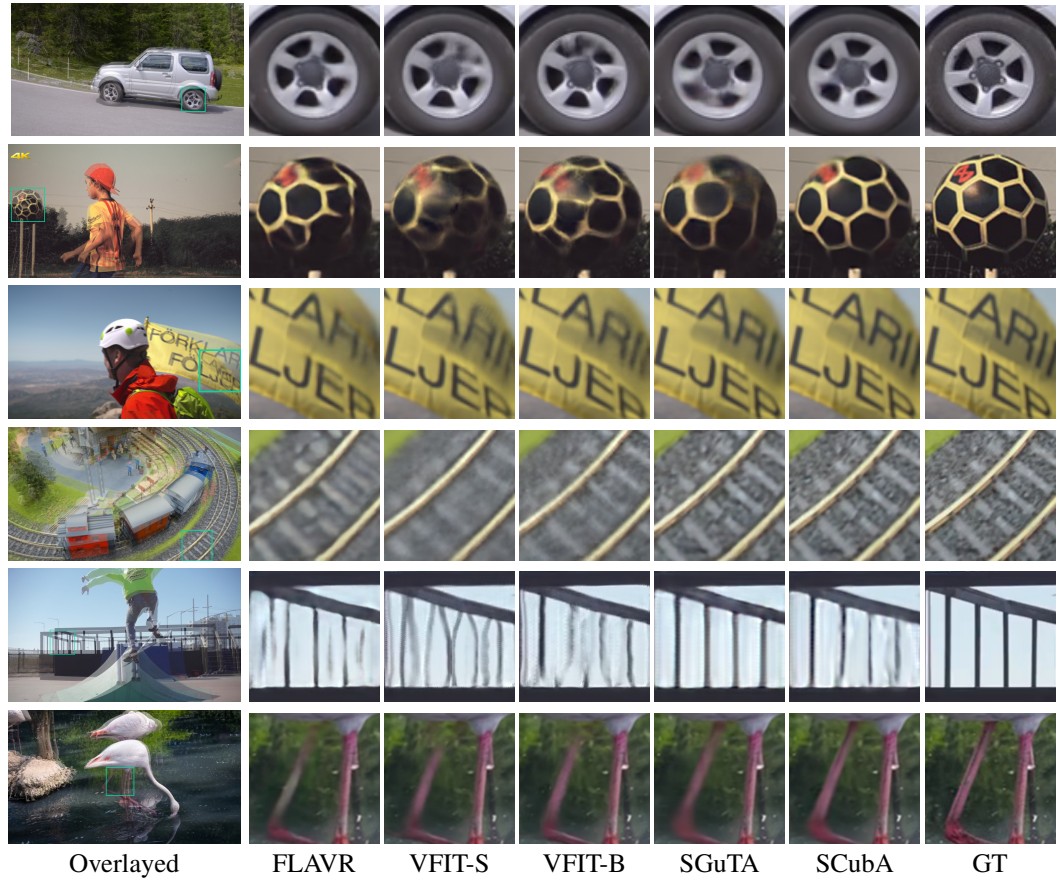

| Overlayed | FLAVR | VFIT-S | VFIT-B | SGuTA | SCubA | GT |

Figure 4: Qualitative comparisons against state-of-the-art VFI methods. Both SGuTA and SCubA outperform others in providing precise motion estimation, clear texture details, handling non-rigid motion and large motion scenarios. Note the rotational position of the wheel and the ball when comparing these methods in the first two rows.

Strategies that satisfy Eq. (11) perform similarly, thus we can use this equation to balance Params, FLOPs, and memory usage. The rationale behind this phenomenon is that the correlation between adjacent patches exhibits redundancy under the Wide-Overlapping Strategy, whereas it manifests sparsity in the Non-Overlapping Strategy. Consequently, the latter requires a lengthier representation to restore the comparable performance. The correlation of Half-Overlapping Strategy lies between the previous two strategies, and the appropriate overlapping region can provide some inductive bias, such as the relative positional information, to MSA. It is worth noting that due to the non-linear nature of Eq. (11), the Half-Overlapping Strategy exhibits a distinct feature of high returns on investment, with lower Params, FLOPs, and memory requirements compared to the average values of the Wide Overlapping Strategy and Non-Overlapping Strategy at the same performance level.

Table 3: Quantitative comparisons on different embedding strategy.

| Embedding Strategy | (Patch Number) $\times D$ | Params (M) | FLOPs (G) | Memory Usage (Gi) | Vimeo-90K |
|---|---|---|---|---|---|
| Wide-Overlapping | $(8 \times 128 \times 128) \times 32$ | 2.99 | 39.60 | 23.78 | 36.03/0.973 |
| Non-Overlapping | $(8 \times 32 \times 32) \times 32$ | 2.99 | 2.99 | 3.15 | 33.64/0.956 |
| Non-Overlapping | $(8 \times 32 \times 32) \times 128$ | 45.24 | 2.99 | 7.75 | 36.04/0.973 |
| Half-Overlapping | $(8 \times 64 \times 64) \times 64$ | 11.53 | 11.53 | 12.38 | 36.08/0.973 |

### 4.3.2 Self-Attention Mechanism

In this section, We first replace all MSA blocks with two layers of 3D ResBlocks [40] to enable a comparative assessment of CNN-based methods with other Transformer-based methods. Subsequently, a thorough evaluation of the performance of different MSAs is conducted. The scrutinized MSA-based methods are enumerated as follows: **1) Baseline** where the MSA modules are all removed from the multi-scale Transformer. **2) STS MSA** [14] and **3) Sep-STS MSA** [14] replaces our MSA modules with STS blocks and Sep-STS blocks [14] respectively. **4) Feature MSA** [21] obtain self-attention from Eq. (8). **5) SGuTA** and **6) SCubA** are the methods proposed in this paper. Besides, **Global MSA** [16] employs Eq. (7) for self-attention, but its performance is unreported due to the excessively high computational complexity (587.93G) and memory requirements. To ensure fairness, all methods are configured with $N_s = 2$ and adopt the half overlapping embedding strategy with $D = 64$. Because the differences between models can be distinguished at the early stages of training, we report performance for all models trained for 20 epochs.

As shown in Tab. 4, on the one hand, compared to the 3D ResBlock method based on CNN, Transformers benefit from their ability to establish long-range dependencies, achieving improved performance with lower Params and FLOPs. On the other hand, compared to the Baseline, Feature MSA only provides a modest improvement in PSNR by 0.06dB, indicating that the self-attention for features has limited benefit for VFI task. STS MSA and Sep-STS MSA show PSNR improvements of 0.37dB and 0.60dB, respectively, with Sep-STS MSA acting similarly to depth-wise separable convolution [34], resulting in a lighter and more efficient STS-MSA. Compared to Feature MSA, SGuTA significantly improves PSNR by 0.54dB, demonstrating the effectiveness of SGuTA in establishing correlations between space and time. SCubA leverages the shifted-cube mechanism to fully exploit the power of local attention, achieving the best performance among all MSAs.

Table 4: Ablation study of different MSA

| Methods | Params (M) | FLOPs (G) | Vimeo-90K |
| --- | --- | --- | --- |
| 3D ResBlock | 17.50 | 57.94 | 34.82/0.966 |
| Baseline | **7.84** | **5.12** | 35.33/0.969 |
| Feature MSA | 8.76 | 22.11 | 35.39/0.970 |
| STS MSA | 11.53 | 37.97 | 35.70/0.972 |
| Sep-STS MSA | 10.61 | 29.76 | 35.91/**0.973** |
| SGuTA | 18.40 | 49.04 | 35.93/**0.973** |
| SCubA | 11.53 | 34.48 | **36.08/0.973** |

Table 5: Ablation study of stage number

| Methods | $N_s$ | Params (M) | FLOPs (G) | Vimeo-90K |
| --- | --- | --- | --- | --- |
| SGuTA | 1 | **9.20** | **24.52** | 35.65/0.971 |
|  | 2 | 18.40 | 49.04 | 37.28/0.979 |
|  | 3 | 27.60 | 73.55 | **37.54/0.980** |
| SCubA | 1 | **5.77** | **17.24** | 36.72/0.976 |
|  | 2 | 11.53 | 34.48 | 37.48/0.980 |
|  | 3 | 17.30 | 51.71 | **38.04/0.981** |

### 4.3.3 Stage

In this section, we explore the impact of the number of cascaded Multi-scale Transformers $N_s$. Due to concerns regarding Params and FLOPs, we only consider the case when $N_s \leq 3$. The results are presented in Tab. 5, where it can be observed that when $N_s = 3$, both SGuTA and SCubA perform the best. Additionally, it is worth noting that when $N_s = 1$, compared to VFIT-S, SCubA achieves a PSNR improvement of 0.24dB while reducing the Params and FLOPs by 23% and 58%, respectively. When $N_s = 2$, compared to VFIT-B, SCubA achieves a PSNR improvement of 0.52dB with 43% of its Params and 40% of the FLOPs.

## 5 Conclusions

In this paper, we employ two different Transformers, SGuTA and SCubA, to the VFI task. SGuTA is designed to establish intrinsic connections between video spatial and temporal information, while SCubA employs a 3D local self-attention mechanism. Both methods are integrated into a multi-stage multi-scale framework. Compared to previous state-of-the-arts, extensive experiments show that our methods achieve the best and second-best performance on multiple benchmarks, and particularly excel in handling large motion and providing accurate motion estimation. Additionally, we summarized the regularity between the patch extraction stride and the length representation when different embedding strategies maintain comparable performance. We will further verify the universality of this regularity, as well as extend our model to multi-frame interpolation in future work.

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
