# OpenReview forum: "Spatially-Guided Temporal Attention (SGuTA) and Shifted-Cube Attention (SCubA) for Video Frame Interpolation"
_NeurIPS.cc/2023/Conference — Submitted to NeurIPS 2023_

### Official Review · Reviewer_RdUT · 2023-06-26

**Soundness:** 2 fair
**Presentation:** 2 fair
**Contribution:** 1 poor
**Rating:** 4
**Confidence:** 5

**Summary:**

This work introduces two attention modules for video frame interpolation that achieve promising results in terms of quantitative and qualitative evaluation. The researchers conducted numerous experiments to provide a comprehensive evaluation of the proposed approach.

**Strengths:**

The proposed attention mechanism enhances the interpolation quality of intermediate frames.


**Weaknesses:**

(1) While this work does not present many theoretical contributions or fundamental insights, it is technically sound and offers improvements and modifications to existing methods. For example, the shift windows attention applied to spatial-temporal aggregation is a relatively simple and incremental approach that builds upon similar ideas explored extensively in previous research.

(2) Table 1/2 should specify the training setups of previous state-of-the-art video frame interpolation methods, as the proposed method uses six input frames. As far as I know, existing models are generally trained on Vimeo-Triplets, which only consist of 51K three-frame samples.

(3) Additionally, the specific contributions of SGuTA are unclear, as SCubA significantly outperforms it with fewer parameters and lower multi-adds. Finally, while many previous works have demonstrated arbitrary interpolation ability, this point does not appear to have been discussed in this work.

**Questions:**


(1) The fairness of comparison in Table 1/2 should be carefully examined to avoid any potential biases or confounding factors that could affect the evaluation and conclusion.

(2) The necessity of SGuTA should be clearly demonstrated with evidence, such as experiments or analyses.

(3) The significance of the contributions of this work should be further discussed, highlighting the novel aspects and potential impact of the proposed approach in the field. This could include a comparison with other state-of-the-art methods and a discussion of potential future research directions.

**Limitations:**

Please see the weakness and question parts.
No significant negative societal impact in this work.

---

> ### Author Rebuttal · Authors · 2023-08-05
>
> We sincerely appreciate your valuable insights and thoughtful evaluation of our work. Here we present our explanation  to the specific points raised during the review process:
>
> 1. The reported results in Tab. 1/2 derived through pretraining the previous state-of-the-art models on the [Vimeo90k Septuplet dataset](http://toflow.csail.mit.edu), an advanced version of Triplets consisting of 65k seven- frame sequences. This was succeeded by conducting inferences on other various datasets.  We have taken every possible measure to ensure the fairness of the comparison. It will be emphasized in the revised manuscript , thank you for bringing this to our attention!
> 2. It is true that SGuTA's performance lags behind SCubA in various aspects. However, it is worth noting that SGuTA surpasses the previous state-of-the-art VFI methods. We share it for anticipating that SGuTA could potentially see further improvements or find applications in other domains, leading to its broader significance. Moreover, a primary objective of this work is to compare the effectiveness of various self-attention mechanisms in the context of VFI tasks (Tab.4). SGuTA and SCubA respectively serve as exemplars of the successful application of global and local self-attention mechanisms to video tasks.
> 3. Thank you for your thoughtful and constructive feedback. In the revised manuscript, we will delve deeper into discussing the significance of the contributions made by this work, which will encompass exploring the potential impact within the field as well as outlining potential directions for future research endeavors.
>
> Again, We are grateful for your time and effort in evaluating our work, which has undoubtedly contributed to its refinement and clarity！

---

> > ### Comment · Reviewer_RdUT · 2023-08-18
> > **Response to the rebuttal**
> >
> > I concur with Reviewer Dfnf regarding the need for clarification on inference fairness. However, even after reviewing the reply, I still have uncertainties regarding the inference settings employed in the proposed methods. In addition, it seems that other existing methods typically utilize a 2-frame setting, such as SepConv.

---

> > > ### Author Response · Authors · 2023-08-20
> > >
> > > Thank you very much for your comments. We have responded to reviewerE6Ri's comments and have offered a comprehensive explanation of our inference method. For more details, please refer to our response.
> > >
> > > Furthermore, our research aligns with the trajectory established by VFIT[1] and FLAVR[2], both of which employ a 4-frame setting. They also compare their methodologies against methods utilizing 2-frame settings, such as SepConv. In fact, we have contemplated the feasibility of adapting 2-frame methods into a 6-frame setting. However, practical implementation proves challenging, as not all methods designed for a 2-frame setting can be seamlessly extended to accommodate a 6-frame setting approach. Even if feasible, varying modification strategies can lead to ambiguity and make it challenging to comprehensively elucidate each method's modifications within the paper.
> > >
> > > We firmly believe that it is essential to break free from the inherent constraints of the traditional 2-frame setting. Imagine the potential if the limitations of memory were not a factor – generating interpolated frames using the entire video as input is an inspiring concept. We hold the conviction that these ideas will become achievable in the near future as hardware infrastructure continues to advance.
> > >
> > > [1]. Zhihao Shi, Xiangyu Xu, Xiaohong Liu, Jun Chen, and Ming-Hsuan Yang. Video frame interpolation transformer, CVPR 2022
> > >
> > > [2]. Tarun Kalluri, Deepak Pathak, Manmohan Chandraker, and Du Tran. Flavr: Flow-agnostic video representations for fast frame interpolation, WACV 2023

---

### Official Review · Reviewer_Dfnf · 2023-07-04

**Soundness:** 2 fair
**Presentation:** 3 good
**Contribution:** 3 good
**Rating:** 5
**Confidence:** 4

**Summary:**

In video tasks, the computational complexity and memory requirements of Transformer is challenging. This paper employ two different Transformers in VFI task resulting in new state-of-the-art results. Furthermore, this paper conduct an analysis of existing embedding strategies, and put forth a novel half-overlapping embedding strategy. The author carried out the experimental analysis carefully.

**Strengths:**

1. I believe the author contributed efficient components to the multi-frame video transformer.
2. The author has open sourced the relevant code.
3. The resulting visuals look good. Overall, the paper has a good impression.

**Weaknesses:**

1. The author did not discuss and compare a series of works on single frame interpolation. CVPR20-SoftSplat, CVPR22-IFRNet, CVPR22-Many-to-many Splatting, ECCV22-RIFE, CVPR23-AMT. This greatly weakens the credibility of model evaluation. Overall, the authors cite very little recent relevant literature.
The model mainly compares the "video frame interpolation transformer", which is not as popular as SoftSplat or RIFE as far as I know, so I hope the author will add more comparison experiments to make the evaluation more solid.
2. The video submitted by the author has only one 720p scene, and there is no other method to compare it.
3. The author has discussed that this model currently does not support multi-frame interpolation.

**Questions:**

1. Please refer to AMT to supplement the results of SNU-FILM.
AMT: All-Pairs Multi-Field Transforms for Efficient Frame Interpolation
2. How did the authors achieve a fair comparison on SNU-FILM? Most of the methods on Table.2 cannot benefit from multi-frame input.
3. How about the actual lentency of the model test, is it still a big improvement? Please report the test results for 720p and 1080p separately.
4. How well does the model perform on 4k related datasets, such as Xiph-4k or XVFI-4k?

Line 39 article->paper
Line 218 we->We

**Limitations:**

Previous video transformers often performed mediocrely at 2k or 4k resolutions, or had a large memory overhead, and it is unclear how well this work in HD scenes.

---

> ### Author Rebuttal · Authors · 2023-08-05
>
> We are sincerely grateful for your time and thoughtful evaluation of our work. Here we present our explanations to the specific points raised during the review process:
>
> 1. First of all, We actually compared our approach with recent relevant works, including CVPR20-AdaCoF, AAAI20-CAIN, WACV23-FLAVR, and CVPR22-VFIT. The reason why we specifically focus on contrasting our method with VFIT is that it was the first to apply the Transformer to VFI task and also concentrated on the Vimeo90k septuplet dataset. Furthermore, based on the reported results in Table 1 for VFIT, our method outperforms SoftSplat on the Vimeo90k septuplet and Davis datasets. However, due to the absence of a publicly available pretrained model trained on the Vimeo90k septuplet dataset, we refrained from reporting its performance on SNU_FILM. Additionally, due to the 100MB attachment size limit, we were unable to upload all comparative results. For more visual results, we kindly refer reviewers to our [webpage](https://www.youtube.com/watch?v=MmX7rkCMVHA). We appreciate your suggestions and will make every effort to incorporate comparisons with the latest methods you mentioned in the revised manuscript.
> 2. Regarding the SNU-FILM dataset, we expanded it into a septuplets dataset for inference. For example, in the "easy" mode, sequence [1, **2**, 3] -> [111, **2**, 3, 4, 5], or [4, 5, 6] -> [2, 3, 4, **5**, 6, 7, 8] (bold indicates ground truth). To ensure fairness, the results presented in Tables 1 and 2 were obtained by retraining on the Vimeo90k septuplets and then inferring on SNU-FILM.
> 3. In terms of our model's inference speed and performance on high-definition datasets, we compared our proposed method against the top two existing multi-frame interpolation methods, FLAVR and VFIT. We initially followed the AMT approach to test their latency at a resolution of 1280 × 720, and subsequently evaluated their performance on the Xiph dataset. All tests were conducted on an NVIDIA RTX 3090 GPU. The results are presented in the table below.
> 4. As observed from the table, as you suspected, our proposed method lags behind FLAVR and VFIT in terms of inference speed. This could be attributed to NVIDIA's specialized acceleration for convolution operations, benefiting FLAVR, and VFIT being a more lightweight Transformer approach inspired by DW convolutions. However, this issue can be mitigated through TensorRT or CUDA acceleration, as our model has relatively lower computational complexity. On the other hand, our SCubA and SGuTA achieve the best and second-best score on the Xiph 2k/4k Benchmark respectively, further demonstrating the superiority of our proposed Transformer method in 2K/4K scenarios, thanks for your suggestion!
>
> We are rather grateful for your feedback, your suggestions have greatly contributed to enhancing this work!
>
> |Methods| FLOPs | PSNR/SSIM(Xiph 2K) | PSNR/SSIM(Xiph 4K) | FPS on 1280 × 720 Image (sec) |
> | ------ | ----- | :----------------: | :----------------: | :---------------------------: |
> | FLAVR  | 42.06 |   36.52 / 0.966    |   33.94 / 0.946    |           **1.24**            |
> | VFIT_S | **7.54** |   37.07 / 0.968    |   34.36 / 0.948    |             1.55              |
> | VFIT_B | 29.08 |   37.36 / 0.969    |   34.57 / 0.949    |             1.86              |
> | SGuTA  | 27.60 |   37.52 / 0.969    |   34.83 / 0.950    |             3.27              |
> | SCubA  | 17.30 | **38.14 / 0.971**  | **35.36 / 0.952**  |             3.68              |

---

> > ### Comment · Reviewer_Dfnf · 2023-08-14
> >
> > Thanks to the author for adding new results, BTW, "FPS on 1280 × 720 Image" should be "Runtime on 1280 × 720 Image".

---

> > > ### Author Response · Authors · 2023-08-14
> > >
> > > Thank you for your kind reminder. We appreciate your time and comments.  This matter will be addressed in the revised manuscript.

---

### Official Review · Reviewer_kW9p · 2023-07-05

**Soundness:** 3 good
**Presentation:** 2 fair
**Contribution:** 3 good
**Rating:** 5
**Confidence:** 2

**Summary:**

This paper aims to address the computational complexity and memory requirements of Transformers to enhance their suitability for the video frame interpolation (VFI) task. The authors propose two novel methods, SCuTA and SGubA, which are integrated into a multi-stage multi-scale framework. SCuTA leverages the correlation between spatial information and temporal sequences, while SGubA incorporates a 3D local self-attention mechanism. Through extensive experiments, the proposed methods demonstrate superior performance in terms of peak signal-to-noise ratio (PSNR) compared to other approaches, while also exhibiting a reduced parameter count.

**Strengths:**

1. Intuitive motivation: The paper provides a clear and intuitive motivation for modifying Transformers to better adapt to the VFI task, addressing the computational complexity and memory requirements.
2. Reasonable design: The introduction of SCuTA and SGubA, which leverage the correlation between spatial information and temporal sequences, is a sensible approach. Additionally, SGubA's utilization of a 3D local self-attention mechanism aligns well with the requirements of the VFI task.
3. Comprehensive experimentation: The paper presents a substantial number of experiments to support the proposed methods. The results consistently demonstrate that the proposed approaches achieve the highest PSNR while also reducing the parameter count, effectively addressing the computational and memory challenges of Transformers in the VFI task.

**Weaknesses:**

1. Lack of clarity in Figure 2: Figures 2d and 2f require further clarification. It is recommended that the authors explain G-MSA and SC-MSA before referring to these figures to enhance reader understanding.
2. Inference time comparison: The paper should provide information on the inference time for all the methods evaluated. This will provide a more comprehensive evaluation of the proposed approaches in terms of both performance and computational efficiency.

**Questions:**

Please refer to weaknesses.

**Limitations:**

The authors do not address the limitations. Please refer to weaknesses.

---

> ### Author Rebuttal · Authors · 2023-08-06
>
>
>
> 1. Thank you for your reminder. We will add explanations for G-MSA and SC-MSA in the revised manuscript.
>
> 2. In terms of our model's inference speed  we compared our proposed method against the top two existing multi-frame interpolation methods, FLAVR and VFIT. We initially followed the AMT approach to test their latency at a resolution of 1280 × 720. By the way, we also evaluate the performance of our model on the Xiph dataset. All tests were conducted on an NVIDIA RTX 3090 GPU. The results are presented in the table below.
>
>    | Methods | FLOPs    | PSNR/SSIM(Xiph 2K) | PSNR/SSIM(Xiph 4K) | FPS on 1280 × 720 Image (sec) |
>    | ------- | -------- | :----------------: | :----------------: | :---------------------------: |
>    | FLAVR   | 42.06    |   36.52 / 0.966    |   33.94 / 0.946    |           **1.24**            |
>    | VFIT_S  | **7.54** |   37.07 / 0.968    |   34.36 / 0.948    |             1.55              |
>    | VFIT_B  | 29.08    |   37.36 / 0.969    |   34.57 / 0.949    |             1.86              |
>    | SGuTA   | 27.60    |   37.52 / 0.969    |   34.83 / 0.950    |             3.27              |
>    | SCubA   | 17.30    | **38.14 / 0.971**  | **35.36 / 0.952**  |             3.68              |
>
>    - It seems that our proposed method lags behind FLAVR and VFIT in terms of inference speed, which could be attributed to NVIDIA's specialized acceleration for convolution operations, benefiting FLAVR, and VFIT being a more lightweight Transformer approach inspired by DW convolutions. However, this issue can be mitigated through TensorRT or CUDA acceleration, as our model has relatively lower computational complexity. Surprisingly, our models achieved the best and second-best scores on the Xiph 2k/4k Benchmark, further demonstrating the superiority of our proposed Transformer method in 2K/4K scenarios.
>
>  We appreciate your time and effort in evaluating this work, which has contributed to its refinement and clarity！

---

> > ### Comment · Reviewer_kW9p · 2023-08-21
> >
> > Thank you for your detailed response. The rebuttal effectively addressed some of my concerns and incorporated crucial experiments. As a result, I will maintain my positive rating.

---

### Official Review · Reviewer_BBtn · 2023-07-06

**Soundness:** 2 fair
**Presentation:** 2 fair
**Contribution:** 2 fair
**Rating:** 4
**Confidence:** 5

**Summary:**

This paper tackles video frame interpolation (VFI). It particularly aims to deploy a transformer architecture for VFI tasks. To address the computational complexity and memory requirements in transformers, it proposes two transformer networks, SGuTa and SCubA. While SGuTa uses the spatial (global) information of video frames to establish temporal correlation, SCubA focuses on local attention. Both methods exhibit linear computational complexities. The authors also introduce a half-overlapping embedding strategy to balance the trade-off between computational complexity and memory usage. Experimental comparisons are presented on several VFI benchmarks.

**Strengths:**

* The work attempts to tackle an important trade-off (performance vs. computation) in using transformers for the VFI task
* The proposed transformers exhibit linear time complexity which makes them easily deployable
* The qualitative results look good
* The ablation studies are thorough and show the effect of the different design choices in the proposed networks
* Code is shared in the supplementary material for reproducibility

**Weaknesses:**

* The motivation of the work is not convincingly justified

   There are several works [2,3] that used a transformer architecture for VFI. However, the authors introduce two new transformer networks without drawing any motivation from the progress that has been made in this line of research. It is also not very clear why the authors need to introduce two different networks in one paper. The introduction part of the paper should be rewritten by positioning the proposed framework in comparison with existing literature and hence filling in the big jump in L38-39.

* The technical novelty of the work is limited

   The proposed modules are heavily copied from other relevant works such as VideoSwin Transformer [1].

   It is also not clear why the proposed multi-head self-attention (MSA) is very different from other MSAs in previous literature. As far as I understand, the key difference is in reshaping the input tensor (HW X TD instead of HWT X D) in SGuTa. The remaining operations are the same as other MSAs. The local attention in ScubA is also similar to the one used in [1].

* The experimental comparisons are limited and unfair

   Experimental results are simply copied (quoted) from previous works. However, the experimental settings the authors used to conduct experiments are very different (6 adjacent frames, different patch sizes) from previous works. Hence, how can the authors convincingly justify that the performance gain is coming from the proposed method and not the different experiment settings?

   Intuitively speaking, using more adjacent frames (6 versus 2 or 4) should provide more context during training, hence, it benefits the proposed method. A fair comparison would follow the common experiment protocol used in previous VFI works.

   The benefit of half-overlapping in Table 3 is not clear. The performance gain is really negligible (0.04dB) compared to the computation overhead (significant increase in memory usage and FLOPs compared to non-overlapping baselines).  how did the authors come to the conclusion in Eq. 11? There has to be a more convincing explanation (proof) than simple observation, otherwise, the claim in L62-64 should be toned down.

   The video presented in the supplementary file only shows one video result for the proposed method. The authors should provide a side-by-side video comparison of their method and competing approaches on different test videos.

* The writing of the paper needs improvement

   The methodology part is very confusing with too many coined terms. It would be better to rewrite this part in a clearer manner.

   It is also not a recommended practice to introduce new acronyms in the title of the paper.

References

[1]. Ze Liu, Jia Ning, Yue Cao, Yixuan Wei, Zheng Zhang, Stephen Lin, and Han Hu. Video swin transformer, CVPR 2022

[2]. Zhihao Shi, Xiangyu Xu, Xiaohong Liu, Jun Chen, and Ming-Hsuan Yang. Video frame interpolation transformer, CVPR 2022

[3]. Liying Lu, Ruizheng Wu, Huaijia Lin, Jiangbo Lu, and Jiaya Jia. Video frame interpolation with transformer, CVPR 2022

**Questions:**

Please refer to the "Weaknesses" section and address the raised concerns carefully.

**Limitations:**

The authors do not discuss the limitations of their work.

---

> ### Author Rebuttal · Authors · 2023-08-06
>
>
>
> Thank you for your thoughtful and constructive feedback. Here, we present our explanations addressing the specific points raised during the review process:
>
> 1. **Concerning the motivation and novelty of the work:**
>    - While it may appear that we did not explicitly reference the progress made in this line of research, we indeed draw inspiration and motivation from various methods. For example, we draw motivation from VFIT [1], which can be interpreted as a space-time separation Transformer inspired by Depth-wise separable convolution. This naturally led us to consider applying a 3D Transformer (SCubA) to the VFI task.
>    - The key distinction of SGuTA extends beyond reshaping the input tensor ( $HW \times TD$ instead of  $HWT \times D$ ). It's important to note that SGuTA's computation for self-attention in each head, denoted as $head_j = V_jsoftmax({{Q}_j^TK_j \over d})$, differs from the conventional MSA, denoted as $head_j =softmax({{Q}_j{K}_j^T \over d})V_j$. Please observe the positions of $Q$, $K$, $V$, and the order of computation. Here, $A_j=softmax({{Q}_j^TK_j \over d})$ is employed to extract spatial similarity information, guiding the generation of the temporal sequence for each pixel position in $V_j \times A_j$. It's due to the cross-dimensional reusability [3] within the video that enables SGuTA's effectiveness (Fig. 1a), also serving as the primary reason for SGuTA's linear computational  relationship with the patch number.
>    - The reason why we introduce two different networks - SGuTA and SCubA - in one paper is that comparing the effectiveness of various self-attention mechanisms in the context of VFI tasks (Tab. 4) is one of the primary objectives of this work. SGuTA and SCubA serve as exemplars of global and local self-attention mechanisms successfully applied to video tasks, sharing the same network architecture but differing in self-attention computation. Discussing them together would not significantly increase the paper length and would facilitate comparing different attention mechanisms.
> 2. **Regarding the network's input and fairness aspects:**
>    - Figure 7 in the FLVAR [2] appendix illustrates that merely increasing the number of input frames does not significantly enhance results and may even lead to degradation. We have validated this finding and found it consistent. Therefore, the choice of the number of input frames should be a consideration for multi-frame interpolation methods, which is why previous works did not use all 6 adjacent frames as input. Moreover, increasing the number of input frames escalates computational complexity, posing challenges for training, particularly with Transformer models. Our approach aims to integrate the 3D Transformer into the VFI task to fully exploit its capabilities. Hence, we diverged from the VFIT [1] or FLVAR [2] approach of using [1, 3, 5, 7]-th frames as input due to our network architecture's temporal downscaling requirement, with the three-scale U-Net necessitating a minimum of 8 frames as input. We have also added two blank frames at the sequence's beginning and end to satisfy this setting.
>    - Due to the 100MB attachment size limit, we could not upload all comparative results. For more visual outcomes, we kindly direct reviewers to our [webpage](https://www.youtube.com/watch?v=MmX7rkCMVHA). We have uploaded comparative results against previous methods on the Xiph 2k/4k and SNU-FILM datasets.
> 3. **Regarding Eq. 11:**
>    - Indeed, Eq. 11 was derived from experimentation and empirical observations. Further rigorous deduction is contemplated as our future work. We also encourage other researchers to participate in discussions on this matter. We will revise the wording in lines 62-64 accordingly. Thank you for your valuable feedback.
>
> We appreciate your suggestions and will make every effort to enhance the clarity and refinement of our writing in the revised manuscript.
>
> References
>
> [1]. Zhihao Shi, Xiangyu Xu, Xiaohong Liu, Jun Chen, and Ming-Hsuan Yang. Video frame interpolation transformer, CVPR 2022
>
> [2]. Tarun Kalluri, Deepak Pathak, Manmohan Chandraker, and Du Tran. Flavr: Flow-agnostic video representations for fast frame interpolation, WACV 2023
>
> [3].Liad Pollak Zuckerman, Eyal Naor, George Pisha, Shai Bagon, and Michal Irani. Across scales and across dimensions: Temporal super-resolution using deep internal learning, ECCV 2020

---

### Official Review · Reviewer_E6Ri · 2023-07-07

**Soundness:** 4 excellent
**Presentation:** 3 good
**Contribution:** 2 fair
**Rating:** 4
**Confidence:** 5

**Summary:**

This paper proposed two types of Transformer for video frame interpolation: Spatially-Guided Temporal Attention (SGuTA) and
Shifted-Cube Attention (SCubA). SGuTa merges the temporal dimension and the embedding dimension during the self-attention process to explore the inherent correlations between the spatial and temporal dimensions. On the other hand, SCubA is focused on reducing the computational complexity and adapts Video Swin Transformer for frame interpolation. Both approaches show significant performance improvements compared with the recent VFI models.

**Strengths:**

1. Good performance. Both of the proposed Transformer models---SGuTA and SCubA---outperform the existing works by a notable margin. As the authors have mentioned, given that the performance of the existing works for VFI is almost plateaued, it seems that the proposed architectures are quite effective.

2. Writing is clear and easy to understand. Also, the figures are neat, and the experiments logically match the paper's claims. The motivations of the proposed approaches are sensible.

**Weaknesses:**

1. Main focus of the paper is a bit confusing.

This paper has two main contributions - SGuTA and SCubA - which are separate contributions, and how to combine these two ideas are not introduced, which makes the paper look incomplete. I would like to suggest three options to remedy this issue:

a) If both SGuTA and SCubA can be combined into a single framework, then this will be the best approach - each module is shown to be effective and can show synergies when combined together.

b) If SGuTA and SCubA cannot be combined, then we should at least discuss when to use which method. It seems like SCubA is quite consistently better than SGuTQ in terms of PSNR/SSIM and also computational complexity - then what is the point of proposing SGuTA? Should we just use SCubA all the time?

c) If neither a) nor b) option is available, then I think we should consider separating SGuTA and SCubA into two distinct papers.

2. Use of more input frames and incapability of generating arbitrary intermediate time step.

The proposed method uses more number of input frames (thus, more information) to predict the middle frame. While methods like QVI [30] also uses more input frames, for truly fair comparison with the other works, I think the authors should also report the performance with triplet-based evaluations.
Also, I have one simple question: it is written in the paper that "septuplets" are used - does this mean that the from the frames 1~7, the input frames are [1, 2, 3, 5, 6, 7] and the GT is 4-th frame? Or, do you use [1, 3, 5, 7]-th frames as the input?

Also, most the existing flow-based works can generate arbitrary time-step intermediate frames, while the proposed SGuTA and SCubA cannot. In my opinion, at least a short discussion regarding this issue is needed.

**Questions:**

Please see Weaknesses.

+ one minor question on the validity of Eq.(11): Is it an empirical finding, or is there any proof for this equation?

**Limitations:**

The limitations of the current work and its potential negative societal impact is not adequately discussed.
The authors are encouraged to think about these issues more seriously and write them, even if the potential societal impact seems to be minimal.

---

> ### Author Rebuttal · Authors · 2023-08-05
>
> Thank you for your thoughtful and constructive feedback.
>
> 1. As you mentioned, SGuTA's performance lags behind SCubA in various aspects. We still chose to include SGuTA in our work for a specific reason.
>    - On one hand, SGuTA demonstrates better performance on various datasets compared to previous methods. By sharing it, we aim to ensure that its potential for improvement is not overlooked.
>    - Furthermore, SGuTA and SCubA share the same network architecture, differing only in the computation of self-attention. Discussing them together would not consume a significant portion of the paper and would facilitate the comparison of various attention mechanisms. We have gained some inspiration for integrating these ideas, and we appreciate your suggestions!
> 2. Regarding the network's input and fairness aspects,
>    - Our model takes the frames [1, 2, 3, 5, 6, 7] as input, and the 4th frame as the ground truth (GT). In fact, we have also padded two blank frames at the beginning and end of the sequence. This padding is necessary because our network architecture considers temporal downsampling, and the three-scale U-Net requires a minimum of 8 frames as input.
>    - Regarding the fairness of the comparison, Figure 7 in the FLVAR [1] appendix illustrates that simply increasing the number of input frames does not significantly improve the results and may even lead to degradation. We have verified this conclusion and found it to be true. Therefore, the number of input frames for the network is an important consideration. In our case, we aim to incorporate the 3D Transformer into the VFI task and fully leverage its capabilities. Therefore, we did not adopt the approach of FLVAR [1] or VFIT [2], using [1, 3, 5, 7]-th frames as input.
>    - Consistent with FLAVR [1] and VFTI_B [2], the comparative results mentioned in the tables were obtained by training on the Vimeo90k Septuplet training dataset and then inferring on other datasets. We have taken every possible measure to ensure the fairness of the comparison.
> 3. Eq. (11) is indeed an empirical finding, and we intend to explore its validation in our future work.
>
> Again, thank you for your valuable input and for aiding us in advancing our research.
>
> References
>
> [1]. Tarun Kalluri, Deepak Pathak, Manmohan Chandraker, and Du Tran. Flavr: Flow-agnostic video representations for fast frame interpolation, WACV 2023
>
> [2]. Zhihao Shi, Xiangyu Xu, Xiaohong Liu, Jun Chen, and Ming-Hsuan Yang. Video frame interpolation transformer, CVPR 2022

---

> > ### Comment · Reviewer_E6Ri · 2023-08-15
> > **Additional questions for clarification**
> >
> > 1. (Comment) Even though SGuTA itself is meaningful in some aspects, I still think it's better to just remove or lower the emphasis on SGuTA  (at least remove it in the title) for better focus of this paper. Currently from the author rebuttal, SGuTA has two meanings: 1) it can potentially show more improvements because it shows better performance than previous works, 2) for more thorough comparison and discussions.
> > However, the main claim of this paper and the effectiveness goes to SCubA, which makes me (and the other potential readers) confused.
> > In my opinion, in order to include SGuTA in this paper, the authors need to derive some general conclusion from SGuTA (even if the derivation is from just a few qualitative examples), which is able to persuade the readers of when to use SGuTA and when to use SCubA.
> > (e.g. SGuTA works well on line patterns, which is because ~~~some intuitive descriptions~~~. In other cases, use SCubA.)
> >
> > 2. Regarding "fairness" of comparison, what I meant was not about the number of frames but about the temporal resolution. [1, 3, 5, 7] --> 4th frame is increasing the temporal resolution by 2. However, using [1, 2, 3, 5, 6, 7] as input frames is not practical in that the [2, 6] frames are required as inputs, which means that the input video already have high temporal resolution and only a single center frame is missing. In such case, how can we interpolate a 30-fps video to make a 60-fps video?

---

> > > ### Comment · Reviewer_Dfnf · 2023-08-15
> > >
> > > The author needs to pay special attention, [1, 2, 3, 5, 6, 7]->4th is not acceptable during inference. Hope to get a direct explanation.

---

> > > ### Author Response · Authors · 2023-08-21
> > >
> > > 1. We sincerely appreciate your constructive feedback and suggestions. In the revised manuscript, we will consider removing "SGuTA" from the title to prevent any confusion for other readers.
> > >
> > >    - In addition, we would like to supplement the novelty of SGuTA with the following clarification: It is important to note that SGuTA's computation for self-attention in each head, denoted as $head_j=V_j softmax({Q^T_jK_j \over d})$ in Eq. (2), differs from the conventional MSA, denoted as $head_j=softmax({{Q}_j{K}_j^T \over d})V_j$. It's important to observe the positions of $Q, K, V$, and the order of computation. Here, The $A_j = softmax({Q^T_jK_j \over d})$ is employed to extract the similarities of spatial features , which in turn play a guiding role in shaping the temporal sequence within $V_jA_j$ process. It's due to the cross-dimensional recurrence [3] within the video that enables SGuTA's effectiveness (Fig. 1a) and this change enables SGuTA the linear computational complexity with respect to number of patches.
> > >
> > >    - We acknowledge your suggestion to include more qualitative results and some general conclusion. For instance: "As illustrated in Fig. 4, SCubA benefits from the Transformer's ability to capture global spatio-temporal dependencies. Compared to other methods, it demonstrates superior adaptability across scenarios involving rigid, non-rigid, linear, and complex nonlinear motions, yielding structurally clearer and less distortions. On the other hand, SGuTA benefits from its capability to capture cross-dimensional recurrence  between the temporal and spatial dimensions [1], resulting in enhanced performance in linear motion scenarios. Despite extreme displacements, SGuTA outperforms FLAVR, VFIT, and SCubA in linear motion scenes. However, it lags behind SCubA in other nonlinear motion scenes such as rotational motion."
> > >
> > > 2. Regarding how to improve frame rates during inference, Our previous explanation may have been insufficiently clear. Here we provide a more detailed distinction between our method and others in terms of inference, training and testing processes :
> > >
> > > - **Inference Process**: Let's consider a video sequence [1, 2, 3, 4, 5, 6, 7, 8...]. FLAVR[2] and VFIT[3] utilize [2, 3, 4, 5] to predict interpolation frames between the 3rd and 4th frame, and [3, 4, 5, 6] to predict interpolation frames between the 4th and 5th frame, our approach employs [1, 2, 3, 4, 5, 6] to predict interpolation frames between the 3rd and 4th frame, and [2, 3, 4, 5, 6, 7] to predict interpolation frames between the 4th and 5th frame.
> > > - **Training or Testing Process**: Considering a sequence [1, 2, 3, 4, 5, 6, 7], where the 4th frame is the ground truth (GT), FLAVR[2] and VFIT[3] discard the 2nd and 6th frames, using [1, 3, 5, 7] as input to predict the 4th frame. In contrast, our method retains the 2nd and 6th frames, using [1, 2, 3, 5, 6, 7] as input to predict the 4th frame. (It's worth noting that during testing on the UCF101 dataset presented in [4], an exception exists where each sequence comprises only 5 frames [1, 2, t, 3, 4]. Consequently, for testing on the UCF101 dataset, both FLAVR and VFIT use the sequence [1, 2, 3, 4] to predict the t-th frame.)
> > >
> > > - If you still find this confusing, we have prepared a demonstration below in hopes of assisting your understanding.
> > >
> > >    - During **inference**:
> > >
> > >      |   Raw time   |    1    |    2    |    3    |    Unknown     |    4    |    5    |    6    |
> > >      | :----------: | :-----: | :-----: | :-----: | :------------: | :-----: | :-----: | :-----: |
> > >      | FLAVR & VFIT | Discard | &check; | &check; | 3.5(Predicted) | &check; | &check; | Discard |
> > >      | SCubA &SGuTA | &check; | &check; | &check; | 3.5(Predicted) | &check; | &check; | &check; |
> > >
> > >    - During **training or testing**：
> > >
> > >      |   Raw time   |    1    |    2    |    3    |    4(GT)     |    5    |    6    |    7    |
> > >      | :----------: | :-----: | :-----: | :-----: | :----------: | :-----: | :-----: | :-----: |
> > >      | FLAVR & VFIT | &check; | Discard | &check; | 4(Predicted) | &check; | Discard | &check; |
> > >      | SCubA &SGuTA | &check; | &check; | &check; | 4(Predicted) | &check; | &check; | &check; |
> > >
> > > We sincerely appreciate your insights and assistance, as they have genuinely helped us enhance our work.
> > >
> > > References
> > >
> > > [1].Liad Pollak Zuckerman, Eyal Naor, George Pisha, Shai Bagon, andMichal Irani. Across Scales & Across Dimensions: Temporal Super-Resolution using Deep Internal Learning, ECCV2020.
> > >
> > > [2].Tarun Kalluri, Deepak Pathak, Manmohan Chandraker, and Du Tran. Flavr: Flow-agnostic video representations for fast frame interpolation, WACV 2023
> > >
> > > [3].Zhihao Shi, Xiangyu Xu, Xiaohong Liu, Jun Chen, and Ming-Hsuan Yang. Video frame interpolation transformer, CVPR 2022
> > >
> > > [4]Soomro Khurram, Zamir Amir, and Shah Mubarak.UCF101: A dataset of 101 human action classes from videos in the wild. CRCV-TR-12-01, 2012.

---

### Author Response · Authors · 2023-08-09
**Concerning the comparative results against previous methods**

1. In terms of our model's inference speed and performance on high-definition datasets, we compared our proposed method against the top two existing multi-frame interpolation methods, FLAVR and VFIT. We initially followed the AMT approach to test their latency at a resolution of 1280 × 720, and subsequently evaluated their performance on the Xiph dataset. All tests were conducted on an NVIDIA RTX 3090 GPU. The results are presented in the table below.

| Methods | FLOPs    | PSNR/SSIM(Xiph 2K) | PSNR/SSIM(Xiph 4K) | FPS on 1280 × 720 Image (sec) |
| ------- | -------- | :----------------: | :----------------: | :---------------------------: |
| FLAVR   | 42.06    |   36.52 / 0.966    |   33.94 / 0.946    |           **1.24**            |
| VFIT_S  | **7.54** |   37.07 / 0.968    |   34.36 / 0.948    |             1.55              |
| VFIT_B  | 29.08    |   37.36 / 0.969    |   34.57 / 0.949    |             1.86              |
| SGuTA   | 27.60    |   37.52 / 0.969    |   34.83 / 0.950    |             3.27              |
| SCubA   | 17.30    | **38.14 / 0.971**  | **35.36 / 0.952**  |             3.68              |

- As observed from the table, our proposed method lags behind FLAVR and VFIT in terms of inference speed. This could be attributed to NVIDIA's specialized acceleration for convolution operations, benefiting FLAVR, and VFIT being a more lightweight Transformer approach inspired by DW convolutions. However, this issue can be mitigated through TensorRT or CUDA acceleration, as our model has relatively lower computational complexity. On the other hand, our SCubA and SGuTA achieve the best and second-best score on the Xiph 2k/4k Benchmark respectively, further demonstrating the superiority of our proposed Transformer method in 2K/4K scenarios!

2. For more visual results, we kindly refer reviewers to our [webpage](https://www.youtube.com/watch?v=MmX7rkCMVHA). We appreciate your suggestions and will make every effort to incorporate comparisons with the latest methods you mentioned in the revised manuscript.

---

### Decision · Program_Chairs · 2023-09-21

**Decision:**

Reject

**Comment:**

This is a borderline submission. The reviewers agree that video frame interpolation is a relevant and well motivated task. Initially, the reviewers outlined several flaws with the empirical validation and questioned the need for both models proposed in the work. These key points were not sufficiently addressed.